# The Mechanism of *Houttuynia cordata* Embryotoxicity Was Explored in Combination with an Experimental Model and Network Pharmacology

**DOI:** 10.3390/toxins15010073

**Published:** 2023-01-13

**Authors:** Yufu Liu, Guodong Yang, Chunqi Yang, Zhuo Shi, Yi Ru, Ningning Shen, Chengrong Xiao, Yuguang Wang, Yue Gao

**Affiliations:** 1School of Pharmacy, Jiangxi University of Traditional Chinese Medicine, Nanchang 330004, China; 2Department of Pharmaceutical Sciences, Beijing Institute of Radiation Medicine, Beijing 100850, China

**Keywords:** embryotoxic, *Houttuynia cordata*, network pharmacology, molecular docking

## Abstract

*Houttuynia cordata* (*H. cordata*) is the most common herb as a food and traditional Chinese medicine. Currently, studies on its toxicity have mainly focused on hepatotoxicity. However, its potential embryotoxicity by long-term exposure is often overlooked. Objective: To investigate the effects of *H. cordata* on embryonic development and its toxicity mechanism by combining network pharmacology, molecular docking, and in vitro experimental methods. Methods: The effects of *H. cordata* on embryos were evaluated. Zebrafish embryos and embryoid bodies were administered to observe the effects of *H. cordata* on embryonic development. Based on network pharmacological analysis, it was found that the main active agents producing toxicity in *H. cordata* were oleanolic acid, lignan, and aristolactam AII. *H. cordata* can affect PI3K-Akt, MAPK, and Ras signaling pathways by regulating targets, such as AKT1, EGFR, CASP3, and IGF-1. RT-PCR and immunohistochemistry results showed that the expression of AKT1 and PI3K in the embryoid body was significantly reduced after drug administration (*p* < 0.05). Conclusions: The results of network pharmacology and in vitro experiments suggest that *H. cordata* may affect embryonic development by influencing the PI3K-Akt signaling pathway.

## 1. Introduction

With the increase in medical skills and the enhancements in people’s lives, the share of the economic market for medicinal and edible plants is gradually increasing. People generally believe that medicinal and edible plants have low toxicity and side effects and pay little attention to the rational use of a medicinal and edible plant with long-term exposure [1], leading to some Chinese medicine causing various adverse reactions. For example, cassia seeds are used as fried tea or as a traditional medicine in Asian countries, and some studies have shown that *cassia* seeds are hepatotoxic and nephrotoxic [2]. When the body is vitamin B6 deficient, consumption of *ginkgo* can cause generalized tonic clonus and mental confusion [3]. With the increase in adverse events of medicinal and edible plants, the evaluation of the toxicity and safety of medicinal and edible plants has increased. Among liver, kidney, heart, and other toxicities, embryotoxicity is the most important, because it may endanger the health and life of offspring in addition to its adverse effects, and once detected [4], it is impossible to recover after discontinuation of the drug.

*Houttuynia cordata (H. cordata)* is a perennial herb in the Saururaceae family [5]. *H. cordata* was first recorded in Ming Yi Bie Lu more than 1500 years ago. It is widely distributed in China, Korea, Japan, India, and other Asian regions, and is important in China [6]. Current pharmacological research indicates that *H. cordata* has anti-mutagenic [7], anti-cancer [8], anti-obesity [9], hepatoprotective [10], anti-viral [11], anti-bacterial [12], anti-inflammatory [13], anti-allergic [14], anti-leukemia [15], chronic sinusitis, and nasal polyps effects [16]. *H. cordata* is also widely consumed as a healthy vegetable in East Asia [17]. For example, in Guizhou Province, its leaves and rhizomes are widely consumed as high-quality agricultural vegetables. The 95% ethanolic extract of *H. cordata* was tested in a single dose and long-term toxicity in rats, and the results showed toxicity to rat liver and kidney [6]. Furthermore, the above-ground part of *H. cordata* and the 95% ethanol extract of the rhizoma *H. cordata* had certain acute toxicity and embryonic development toxicity to zebrafish adults and embryos. *H. cordata* is often used as a way to eat stews and boiled dishes, and the method of water extract is closest to daily consumption [6]. The study on the components of water extract of *H. cordata* has great practical significance. However, the embryo safety of water extract *H. cordata* remains unknown.

In 2007, the British pharmacologist Hopkins formulated the idea of network pharmacology [18]. Network pharmacology is based on systems biology and pleiotropic pharmacology with the integration of computational biology and network analysis [19]. It illustrates the complete and systematic character of drug-target-disease interactions from a perspective of multiple components, targets, and channels, which is in line with the holistic view of Chinese medicine [20]. Hence, network pharmacology shows great potential in evaluating the efficacy and toxicity of TCM. In this study, we first used zebrafish and embryoid bodies to verify the embryotoxic effects of *H. cordata* more sensitively. We then used a network pharmacology method to unravel the mode of action of *H. cordata* in terms of embryotoxicity. Simultaneous validation of core targets was performed using network pharmacology and molecular docking. This study will facilitate the safety assessment of *H. cordata* and provide information for the clinical use of *H. cordata*. 

## 2. Results

### 2.1. Zebrafish Mortality Effects

For the assessment of the effect of *H. cordata* on embryonic mortality in zebrafish, Zebrafish embryos at 0 hpf were exposed to 2, 2.5, 3, 3.5, and 4 mg/mL of each tested compound until 120 hpf. The first row of Figure 1 shows the developmental process of zebrafish, where 0–24 hpf belongs to the embryonic maturation stage, 48–72 hpf is the adult stage of zebrafish hatching, and after 120 hpf all organs of zebrafish have been developed. At present, there are four criteria to evaluate zebrafish mortality. The first one is zebrafish embryo coagulation, the second one is zebrafish missing body segments, the third is when the zebrafish is not detached from the tail, and the fourth is when the zebrafish has no heartbeat; then, it is determined to be dead [21]. The results of this study showed that the mortality of zebrafish at 24, 48, and 72 hpf gradually exhibited embryonic coagulation with increasing doses of *H. cordata* drugs. At 96 and 124 hpf, zebrafish were administered with 3.5 and 4 mg/mL of *H. cordata* solution, died, and had curved spines. The red arrows in Figure 1 indicate dead zebrafish and the black arrows indicate surviving zebrafish.

The influence of *H. cordata* on the survival percentage of 24, 48, 72, 96, and 120 hpf is shown in Figure 2A. At 24, 48, and 96 hpf, concentrations lower than 2 mg/mL did not significantly induce the mortality of zebrafish embryos compared with the control group. In contrast, compared with the control group, 2 mg/mL of *H. cordata* markedly decreased the survival rate of zebrafish embryos. At 120 hpf, all embryos of 3 mg/mL *H. cordata* zebrafish died. The LC_50_ of *H. cordata* at 120 hpf were 2052 μg/mL.

### 2.2. Zebrafish Hatching Rates and Morphological Deformities

The hatching rate is a vital process in zebrafish embryo development, and the hatching rate has been used as an indicator to assess the toxicity of zebrafish embryo development. Exposure to 2 mg/mL of *H. cordata* markedly decreases the hatching rate of zebrafish embryos (Figure 2B). The main target organ of *H. cordata* toxicity to zebrafish was the cardiovascular system compared to the control, with indirect toxic effects on the liver and intestine. The main manifestations were pericardial edema, abnormal heart rhythm, slow blood flow, loss of circulation, delayed yolk sac absorption (abnormal liver function), small liver, and abnormal intestine (Figure 3A,B). These outcomes indicated that the inhibitory effect of *H. cordata* on zebrafish embryonic development was relatively stronger.

### 2.3. Embryoid Body Morphogenesis Is Impacted by H. cordata

EBs were explanted in *H. cordata* solutions at concentrations of 150, 250, and 350 μg/mL, and morphological parameters (diameter) were evaluated over 4 days. Unfavorable morphogenetic effects were observed when EBs were exposed to concentrations of 150 μg/mL or more. EBs were smaller (i.e., smaller relative diameter) than the control when exposed to 150 μg/mL. When treated at 350 μg/mL, cells did not survive forming EBs. To further refine the concentrate–response relationship, EBs were exposed to 150, 250, and 350 μg/mL of *H. cordata* (Figure 4A). This indicated that *H. cordata* was significantly embryotoxic (Figure 4B).

### 2.4. Active Compounds and Targets of H. cordata

We found a sum of 189 drug-related components from the TCMSP database and the existing literature, and then screened them by the criteria of DL ≥ 0.18 and OB ≥ 30% to obtain 51 active components, and the specific target information is shown in Table 1. The 2D SDF structures of these 51 compounds were uploaded to search for prospective toxic targets from the Swiss target prediction database, and 1034 component-related targets, and finally the repeat targets were eliminated by comparison and correction of the UniProt database. A sum of 252 gene targets was acquired, as shown in Figure 5A.

### 2.5. Gene Targets in Embryotoxicity

In the present study, 1500 gene targets associated with embryotoxicity were derived from the Gene Cards database, and an additional 160 gene targets were derived from the OMIM database, of which 98 were identical. After deleting the duplicated genes, the number of total embryotoxicity-associated gene targets was 1562. Next, 126 mutually interacting gene targets were acquired by crossing over disease-related gene targets and drug-related gene targets (Figure 5B).

### 2.6. PPI Network Analysis

The 126 phase-crossing gene targets were presented to the STRING database, and then the data gained from this platform were submitted to Cytoscape 3.7.2 for the visualization and topological analysis of the PPI network consisting of 126 nodes and 1198 edges. The topological properties are analyzed using CytoNCA plug-in for the intersecting gene targets with median values of 0.002626815, 0.496971725, and 15 for BC, CC, and DC, respectively. The values of 49 gene targets were above the median, but there were still too many targets. The median of degree, BC, and CC was used to extract 17 targets. These genes were considered to be important genes for *H. cordata* embryotoxicity. From these 17 genes, the degree method was used to predict the embryotoxicity genes. Among the 17 genes, the AKT1 degree value was the highest, indicating that this gene had the greatest correlation with embryotoxicity (Figure 5C).

### 2.7. GO and KEGG Pathway Enrichment Analyses

Gene Ontology (GO) analysis and Kyoto Encyclopedia of Genes and Genomes (KEGG) analysis are essential sub-analyses in network pharmacology to analyze the function of genes and the associated enriched pathways. Enrichment analysis was performed by GO and KEGG. 126 cross-targets were entered into the Metascape system for GO enrichment analysis and pathway enrichment analysis. The biological processes (BP), molecular functions (MF), and cellular components (CC) associated with GO results revealed the functions of these underlying targets. Altogether, 2766 significantly enriched GO entries were derived from Metascape, including 1642 BP, 723 MF, and 401 CC entries, based on *p* < 0.01. Out of 1642 BP GO terms, the most enriched BP terms were positive regulation of response to the hormone, cellular response to hormone stimulus, and cellular response to the organic cyclic compound. The most important terms among the 723 MF GO terms related to embryotoxicity were lipid binding, alcohol groups as acceptors, and kinase activity (Figure 6). Among 401 CC GO terms associated with embryotoxicity, secretory granule lumen, side of vesicle lumen, and cytoplasmic vesicle lumen were included in the top 10 entries. There were 141 pathways enriched according to *p* < 0.01. The pathways with the largest enrichment in terms of levels included PI3K-Akt, EGFR, and MAPK signaling pathways. According to the *p* value as well as the number of hit genes, the top 10 GO entries and the top 20 KEGG paths were picked (Figure 7).

### 2.8. Network Construction

To further illuminate the embryotoxic mechanism of action of *H. cordata*, in this part, a systematic and comprehensive bioinformatics network of herbal medicine–active compound–central target–pathway–embryotoxicity network (C-E-H-P-E) was constructed by 51 active compounds, 126 potential gene targets, and 20 KEGG signaling pathways. The complex network consisted of 211 nodes and 2572 edges, showing complex interactions between various compounds, targets, and pathways. In the bioinformatics network, purple represents herb; blue indicates bioactive compounds from *H. cordata*; yellow stand for the target genes; green represents signal pathway; red is a disease. One gene target corresponds to multiple compounds, and one potential active compound corresponds to multiple targets. It could be seen that the mechanism of *H. cordata*’s embryonic toxicity is characterized by several components, multiple targets, and various mechanisms. The details of the Chinese medicines-effective compounds-hub targets-pathways embryotoxicity network are shown in Figure 8. C-E-H-P-E bioinformatics network showed that many target genes of *H. cordata* were strongly associated with signaling pathways, as shown in Figure 9.

### 2.9. Molecular Docking Verification, Immunohistochemical and qRT-PCR

In molecular docking, the ligand binds to one or more amino acid residues in the active pocket via hydrogen bonding and participates in the process of conformational change and energy complementation. The binding site, binding fraction, and root-mean-square deviation (RMSD) value can visualize the interaction and stability of the docking model. Here, we constructed AKT1-based protein receptors, which are the core targets of PPI. Based on network pharmacology, three highly active compounds from *H. cordata* were screened for molecular docking. Before performing molecular docking of *H. cordata* components, we first verified the reliability of the molecular docking process. Since the crystal structure of the AKT1 (4EJN) target protein contains the original ligand of 0R4. we extracted the ligand molecule from the complex and then docked it into the binding pocket of the complex using LibDock. The RMSD of the docked ligand from the ligand conformation in the original crystal structure was calculated to be 1.3447, which is less than 2.0, indicating that the docking procedure has good accuracy for this target. Subsequently, we performed molecular docking with three active ingredients, luteolin, Aristolochia lactam AII, and oleanolic acid, as ligands and key target proteins as receptors. The docking results showed that the docking scores of all three active compounds with the core target AKT1 were greater than 100, indicating a relatively stable binding between ligand and receptor (Table 2, Figure 10).

Figure 11 shows the binding pattern of the AKT1 target protein to its corresponding original ligand and the three *H. cordata* active compounds mentioned above. A high similarity was observed between the active compounds and the original ligands of the proteins, which occupy almost the same active site. The binding pattern between oleanolic acid and AKT1 is shown in Figure 11A. The residue Arg273 interacts with the hydroxyl group, forming a stable hydrogen bond. Figure 11B shows the binding mode of luteolin to the AKT1 active site. Three amino acid residues form hydrogen bonding interactions with luteolin, where the carbonyl group on this compound is attached to residues ASN204 and SER205 via two hydrogen bonds, respectively, and the hydroxyl group is attached to residue THR211. Figure 11C shows the binding pattern between Aristolochia lactam AII and AKT1, where the Ser205 in AKT1 forms a stable hydrogen bond interaction with the carbonyl group. Immunohistochemistry and qRT-PCR were performed at the gene and protein levels, respectively, to detect the effect of Cichorium on AKT1 PI3K expression, and the results showed that *H. cordata* significantly inhibited the expression of PI3K AKT1 gene (Figure 12)

## 3. Discussion

In the current research, zebrafish and embryoid bodies were used to assess embryotoxicity and network pharmacology was used to further investigate the mechanisms of *H. cordata* embryotoxicity. *H. cordata* is marketed as a medicinal and edible plant with many health benefits, e.g., hyperlipidemic [9], neurotoxicity [22], and pulmonary fibrosis [23]. Nevertheless, the embryotoxicity of *H. cordata* itself is not well studied and confirmed. The beneficial or harmful effects of a particular drug on embryotoxicity depend on the dose or concentration of exposure. Therefore, it is critical to determine the dose–response relationship for the adverse effects of *H. cordata* that may be consumed by women of childbearing age. The embryotoxicity of *H. cordata* was evaluated by in vitro embryoid body experiment and in vivo experiment of zebrafish. The possible mechanism was further explored through network pharmacology. The information gained from this study is valuable for studies in humans.

First, in an in vivo experiment, we investigated that the mortality of zebrafish embryos is enhanced in a concentration and a temporal-dependent way. The LC_50_ value of *H. cordata* at 120 hpf was 2052 μg/mL. In conclusion, *H. cordata* had more detrimental effects on embryo survival. The influence of *H. cordata* on the hatching rate of zebrafish embryos was investigated, and it was found that 2 mg/mL *of H. cordata* can significantly decrease the hatching rate of zebrafish embryos. Meanwhile, there was a dose-dependent effect of *H. cordata* on the hatchability of zebrafish embryos. In addition, the experimental results showed that organ toxicity mainly acted on the heart. In in vitro studies, Ips suspension can spontaneously form embryoid body (EBs), which contains endodermal, mesodermal, and exodermal tissues, and can simulate the dialogue and mutual induction process among multiple layers in vivo. When EBs were treated with *H. cordata* at 150 μg/mL and higher doses, adverse morphogenetic effects were observed. The second day of EBs formation corresponds to the onset of embryonic development of implantation and protointestinal formation [24]. On day 4 of the differentiation of EBs, differentially differentiated cell lines begin to appear. On day 6 of EBs differentiation (corresponding to the early stages of organogenesis), the germinal bodies gradually develop into vacuolated embryos. From the present study, it was found that *H. cordata* was significantly toxic to different cell lines from protointestinal embryo formation to differentiation.

Most of the reports on the toxicity of *H. cordata* are focused on hepatic and renal toxicity, and there are few reports on embryotoxicity due to insufficient model test sensitivity. We used zebrafish and embryoid bodies assay to verify embryotoxicity, which was more sensitive than traditional animal experiments. It can be determined that *H. cordata* has a potential embryotoxic risk. Next, the possible mechanism of *H. cordata* embryological toxicity was systematically studied by network pharmacology and molecular docking analysis. The composition-target-toxicity network of *H. cordata* on embryotoxicity was analyzed and constructed. 52 main active ingredients of *H. cordata* and 126 common targets of embryoid bodies, including oleanolic acid, Luteolin, and Aristolochia lactam AII. Studies have shown that oleanolic acid can cause liver damage when taken in small doses over a long period or in large doses over a short time, Cholestasis is the main feature of liver injury [25]. Liver damage was also observed in zebrafish target organ toxicity. In vitro studies showed that administration of lignocaine to lymphoblastoid TK6 cells and a newly developed TK6-derived cell line was found to induce cytotoxicity, apoptosis, DNA damage, and chromosomal damage in a concentration-dependent manner. The cytotoxicity and genotoxicity induced by lignocaine were determined by high-throughput micronucleus assay [26]. It has been shown that hesperidin contains Aristolochia lactam BII, AII, and AIIIa, among which Aristolochia lactam AII is the most cytotoxic, and Aristolochia lactam AII is also toxic to zebrafish embryos [7]. Thus, consistent with previous studies, these major components of *H. cordata* play an important role in embryotoxicity

Using integrated network analysis, we found that *H. cordata* acts on several targets through various signaling pathways, mainly AKT1, EGFR, IGF-1, and CASP3, and that these potential core target genes may play a crucial role in embryotoxicity. EGFR is a specific receptor of the tyrosine kinase receptor family that acts on its placental target cells (trophoblast cells). Changes in the biological activity of EGFR can affect fetal development [27]. EGFR inhibits trophoblast proliferation and induces abortion [28]. At the same time, studies have shown that inhibition of EGFR inhibits angiogenesis [29]. The pro-apoptotic protein Caspase-3 is part of a family of caspases, a group of proteins that have an important role in the mechanism of apoptosis Caspase-3 is the only member of caspases found to be expressed in fetal placental tissue. Once caspase-3 is activated, apoptosis will be irreversible, and the activation of the apoptotic pathway will promote a large amount of apoptosis of embryonic villus cells, causing the embryo to stop developing [30]. The serine/threonine protein kinase Akt is a known oncogene that regulates many processes, including metabolism, proliferation, cell survival, and growth [31]. Studies have demonstrated that dibutyl phthalate is toxic to embryos by regulating AKT1 [31]. Insulin-like growth factor 1 (IGF-1) has long been known to play a role in promoting growth and development, especially in regulating embryonic growth and development [32]. IGF-1 promotes protein synthesis, glucose uptake, cell division, and proliferation. Studies have shown that when IGF-1 can be used as a marker of abortion [33]. It was also found that IGF-1 could be regulated to inhibit growth and development [34]. In conclusion, the above targets were strongly associated with the occurrence of embryotoxicity.

Within our research, KEGG enrichment analysis revealed 126 common targets mainly associated with PI3K-Akt, Ras, and MAPK signaling pathways. *H. cordata* may produce embryotoxicity through the PI3K-Akt signaling pathway, and studies suggest that PI3K-Akt is an important trophoblast regulatory signal in humans [35]. Embryonic toxicity occurs when the PI3K-Akt signaling pathway is inhibited, thereby inhibiting trophoblast cell migration. Repression of the PI3K-Akt signaling pathway leads to decreased phosphorylation of apoptotic proteins and decreased binding to apoptotic molecules. The increase of proapoptotic protein promotes the production of embryonic toxicity. MAPK and Ras signaling pathways participate in the regulation of basic biological activities of cells, including proliferation, apoptosis, differentiation, and senescence. MAPK cascade reaction is Ras dependent, and its abnormal activation can cause an imbalance of cell proliferation/apoptosis. The study found that inhibition of the MAPK pathway leads to abnormal intestinal neurodevelopment in zebrafish. Intestinal neurodevelopment is an important process of embryonic development [36]. Our data support embryotoxicity by inhibiting differentiation or cell proliferation. In conclusion, our data produce embryotoxicity by inhibiting differentiation or cell proliferation.

Our study further analyzed the BP from GO to identify the relationship between underlying targets and embryotoxicity. Based on what we have analyzed, the most abundant BP were cellular response to hormone stimulus, cellular response to organic cyclic compound, and cellular response to nitrogen compound. Nitrogen acquisition and utilization are critical in embryonic development [37]. This provides another insight into the mechanism by which *H. cordata* produces embryotoxicity.

Molecular docking is the most commonly used method to assess component–target interactions. Luteolin, Aristolochia lactam AII, and oleanolic acid bind most stably to AKT1, both with docking fractions < 100. AKT1, EGFR, CASP3, and IGF-1 showed a strong association with all three core components with docking scores < 80. Major components of *H. cordata* can bind more stably to core targets of embryotoxicity. Further, immunohistochemistry and qRT-PCR experiments were performed on the highest-scoring target AKT1 and PI3K, and the results showed that Ichthyophonus had an inhibitory effect on AKT1 and PI3K. Our data suggest that PI3K-Akt may be the most central signaling pathway for the production of embryotoxicity in *H. cordata*.

## 4. Conclusions

In this study, we investigated the effects of *H. cordata* on the development of zebrafish embryos and EBs with iPSCs grown in three dimensions and zebrafish embryos as the model organism. The results showed that the effect of *H. cordata* embryos was evidenced by the reduced diameter of EBs, reduced hatchability, mortality, and teratogenicity of zebrafish embryos, and target organ toxicity. Network pharmacology identified 51 key active compounds and 126 genes targeted by embryotoxicity in *H. cordata*. By integration of network pharmacology, three core targets from a complex network (AKT1, EGFR, CASP3, and IGF-1) were predicted as key targets for embryo production in *H. cordata*. In addition, we discovered that the PI3K-Akt signaling pathway may be central in generating the embryotoxic system. The results of molecular docking further showed that *H. cordata* showed a very good affinity for the core targets, and the immunohistochemical and qRT-PCR results demonstrated the inhibition of AKT1 and PI3K by *H. cordata*. In summary, this research offers a clear basis for the generation of embryotoxicity in *H. cordata* and presents a comprehensive and novel approach to the search for active ingredients, nuclear target genes, and underlying mechanisms of herbal medicines. Pregnant women should eat with caution or watch the amount they consume.

## 5. Materials and Methods

### 5.1. Embryo Toxicity Test of Zebrafish

#### 5.1.1. Test Compounds

*H. cordata* (XEB181) from Lishui, Zhejiang Province was obtained from Huamiao Pharmaceutical Co., Ltd. (Beijing, China), and the drug was extracted with water (5 drinks g/extract g) from the Chinese Academy of Traditional Chinese Medicine. The quality of *H. cordata* was controlled by efficient liquid chromatography (HPLC) (Appendix A). A mother liquor of 5 mg/mL was prepared by using standard dilution water for the powder of *H. cordata* and treated with ultrasonic waves for 10 min before use.

#### 5.1.2. Zebrafish Maintenance and Embryo Collection

AB–Adult feral type zebrafish were fed in fish farming water at 28 °C (water quality: 1 L counter seep water with 200 mg fast dissolving sea salt, conductivity 450~550 μS/cm; PH 6.5~8.5; hardness 50~100 mg/L CaCO_3_), provided by the fish farming center of Huante Biotechnology Company Limited. The laboratory animal permit number is SYXK (Zhejiang) 2012-0171, and the raising management satisfies the international AAALAC requirements for approval (certification number: 001458). For breeding, males and females are separated in a spawning box (having a glass partition) in a proportion of 1:1. The males and females are then mixed and spawned during the early morning hours. Normally developing embryos are captured for lab studies [38].

#### 5.1.3. Exposure of Embryos to Test Compounds

The embryo acute toxicity test was conducted following the methods prescribed by the Organization for Economic Cooperation and Development (OECD) [21,39]. Experimental concentrations of *H. cordata* (2, 2.5, 3, 3.5, and 4 mg/mL) were determined by preliminary tests. Embryos were randomly collected and transferred to 6-well plates after 0 h post-fertilization (hpf), with 30 embryos and 3 mL of *H. cordata* solution deposited in per well. Each concentration and control was replicated 3 times. All test solutions were changed every 24 h. Dead embryos were cleaned up in and timely recorded.

#### 5.1.4. Acute Toxicity Assessment

Each treatment group was watched for developmental toxicity in the exposure period with light microscopy (SZX7, OLYMPUS, Tokyo, Japan). Calculation of zebrafish mortality at 24, 48, 72, 96, 120 hpf, and 72 hpf hatch [40]. At 24, 48, 72, 96, and 120 h after treatment, the development of embryos at different concentrations of liquid was observed by microscope, and the teratogenic site was recorded by CCD camera photo (VertA1, Shanghai Tusen Vision Technology Co., Ltd., Shanghai, China).

### 5.2. Embryoid Body (EBs) Assay

#### 5.2.1. Cell Culture and Embryoid Body Generation

Human-induced pluripotent stem cells (hiPSCs) were purchased from Beijing Saibei Biotechnology Company, placed on 0.1% gelatin (cat. #G2500, Sigma-Aldrich, St. Louis, MO, USA)-coated tissue culture plates, and cells were maintained in the mTeSRTM1 complete medium (85850, stem cell company, Shanghai, China). For embryoid body (EBs) generation, ihPSCs were dissociated to single cells with PSCeasy^®^ Human Pluripotent stem cell Digestive Fluid (CA3001500, Beijing Saibei Biotechnology Company, Beijing, China) and seeded onto ultra-low adhesion U-bottom 96-well plates at a density of 1 × 10^4^ cells/mL in 100 μL EBs differentiation medium (ESC medium without LIF). Embryoid bodies were obtained the next day [41].

#### 5.2.2. Exposure of the Embryoid Body to Test Compounds

The experimental concentrations of *H. cordata* (150, 250, and 350 μg/mL) were determined with the preliminary test. A blank group and a positive control group of mifepristone were also set. Select EBs with uniform appearance and size for culture, and start adding medicine two days later. EBs were cultured for up to 4 days in an incubator with 5% CO_2_ at 37 °C in humidified air. Media were replaced every other day. The survival and integrity of EBs were monitored daily, and the diameter of EBs was measured using a live cell imager. Chemical treatment was considered to have an unfavorable morphogenetic influence when EBs displayed a mean relative diameter that was more than 18% lower than that of the control group [42,43]. Each concentration and control was replicated 3 times.

### 5.3. Exposure of the Embryoid Body to Test Compounds

The results were calculated using GraphPad Prism 8.0 software (La Jolla, CA, USA). The measured data were statistically analyzed using one-way ANOVA and the data were analyzed by one-way ANOVA and chi-square test. The results were analyzed using the mean ± standard deviation (x ± s) The results were expressed as mean ± standard deviation (x ± s), and the differences were considered statistically significant at *p* < 0.05.

### 5.4. Network Pharmacology

#### 5.4.1. Bioactive Ingredients and Targets Screening of *H. cordata*

Through the Traditional Chinese Medicine Systems Pharmacology Database and Analysis Platform (TCMSP, https://tcmspw.com/tcmsp.php, accessed on 22 June 2022), we searched for the effective ingredients of herbal medicine and added them to the published literature. We carried out preliminary filtering of active ingredients to acquire the active compounds and their protein targets, where oral bioavailability (OB) ≥ 30% and drug similarity (DL) ≥ 0.18% were set as criteria. The SwissTarget Prediction platform (http://www.swisstargetprediction.ch/, accessed on 22 June 2022) was then used to predict potential gene targets. Following the screening, the UniProt database (Universal Protein Resource, https://www.uniprot.org/, (accessed on 22 June 2022) was also used to standardize the targets of the proteins in which the compounds act to obtain more comprehensive target information for normalizing protein target information [44].

#### 5.4.2. Toxicity Targets Acquisition of Embryotoxicity and Filtering Intersecting Targets

We mined the GeneCards database (https://www.genecards.org/, accessed on 1 July 2022) and OMIM database (https://omim.org/, accessed on 1 July 2022) for gene targets associated with embryotoxicity using the keyword “embryotoxicity”. Targets related to embryotoxicity were acquired by combining targets from the two databases and deleting duplicates. These co-targets were then filtered using Venny 2.1 (http://bioinformatics.psb.ugent.be/webtools/Venn/, accessed on 1 July 2022) and classified as possible targets for *H. cordata* in embryotoxicity [45].

#### 5.4.3. Protein-Protein Interaction (PPI) Network Construction

PPI networks were structured by presenting common disease–drug targets to a search vehicle for the retrieval of interacting genes (STRING, https://string-db.org/, accessed on 20 July 2022). The organism type was assigned as “Homo sapiens” and the confidence score of the degree of correlation was set to ≥0.4. After hiding the unlinked nodes, the outcome is exported in Tab-separated value (TSV) format. Then, Cytoscape v3.7.2 was employed to visually process the PPI data to filter the kernel targets with the variables of betweenness centrality (BC), proximity centrality (CC), and degree centrality (DC)[46].

#### 5.4.4. Gene Ontology (GO) and Kyoto Encyclopedia of Genes and Genomes (KEGG) Pathway Enrichment Analyses

Previously gained embryotoxicity targets of *H. cordata* were imported into the Metascape system (http://metascape.org/, accessed on 20 July 2022). We selected “Homo sapiens” as the species and a significance level was set at *p* < 0.01 to complete GO functional analysis and KEGG pathway analysis. GO enrichment was performed to cover three aspects of biology: cellular components, molecular functions, and biological processes. KEGG enrichment analysis can also suggest the biological mechanisms of drug action in humans and the pathways involved in their regulation [46].

#### 5.4.5. Network Construction

Network visualization software Cytoscape 3.7.2 was utilized to build the network, a network visualization software to intuitively display the compound–target–pathway relationship between *H. cordata* and Embryo toxicity [47].

#### 5.4.6. Molecular Docking

In the current study, we selected the three active ingredients predicted by web pharmacology as ligand compounds and selected AKT1, which scored the highest in the web pharmacology search filter, as the receptor protein and downloaded the crystal structure of the protein. Molecular docking was performed using Discovery Studio, and the active site was determined using the ligand expansion method, i.e., the location of the ligand was taken as the center and then extended outward to a certain range, generally with a radius of 9 Å, and the receptor residues within this range constituted the relevant active site. Before the molecular docking of the three active ingredients, the accuracy and reliability of the docking process should be evaluated. The target protein complex containing the original ligand is first obtained from the RCSB PDB database (http://www.rcsb.org/, accessed on 20 July 2022), after which the original ligand is abstracted and then docked into the binding pocket of the complex and the root-mean-square deviation (RMSD) of the docked ligand conformation from the ligand conformation in the original crystal structure is calculated. The method is generally considered reliable when the RMSD is less than or equal to 2.0.

#### 5.4.7. Immunohistochemistry

Well-shaped EBs were selected and fixed in 4% paraformaldehyde-phosphate buffer salt solution fixative for more than 24 h. EBs were sedimented in sucrose solution, embedded in OCT, and sliced with a frozen section machine (CM1950, Leica, Wetzlar, Germany) at a thickness of approximately 12–15 μm. Subsequent frozen sections were fixed and antigenically repaired with EDTA antigen repair, endogenous peroxidase was blocked with 1% H_2_O_2_ for 10 min and 5% normal BSA to block nonspecific binding for 30 min. Sections were incubated with phosphate-Akt1 (1:200, servicebio, Wuhan, China) and phosphate-PI3K (1:100, abcam, Shanghai, China) overnight at 4 °C while PBS was added dropwise to each group of samples as a negative control. Subsequently, sections were incubated with goat anti-rabbit enzyme-labeled secondary antibody for 2 h at room temperature and diaminobenzidine for visualization. Finally, these sections were observed and analyzed using a Cytation 5 microscope (BioTek Instruments, Inc., headquartered in Winooski, VT, USA).

#### 5.4.8. RNA Isolation and Quantitative Real-Time Polymerase Chain Reaction (qRT-PCR)

Total sample RNA was extracted using Trizolszol up (TransGen Biotech, Beijing, China) and reverse transcribed into cDNA using the TransScript One-Step gDNA Removal and cDNA synthesis SuperMix (TransGen Biotech, Beijing, China) according to the manufacturer’s guidelines, RT-PCR was performed on an ABI Step One system (Thermo, Shanghai, China) using the TransStrart Green QpcrSuperMix kit (TransGen Biotech, China). The specific primers used for RT-PCR are as follows. PI3K (F: CCACGACCATCATCAGGTGAA;R:CCTCACGGAGGCATTCTAAAGT), AKT1 (F: TCCTCCTCAAGAATGATGGCA;R:GTGCGTTCGATGACAGTGGT), GAPDH (F:GAGTCAACGGATTTGGTCGT; R: TTGATTTTGGAGGGATCTCG). GAPDH was used as a normalization control. The relative RNA expression of each gene was analyzed by the 2^−ΔΔ^CTmethod.

## Figures and Tables

**Figure 1 toxins-15-00073-f001:**
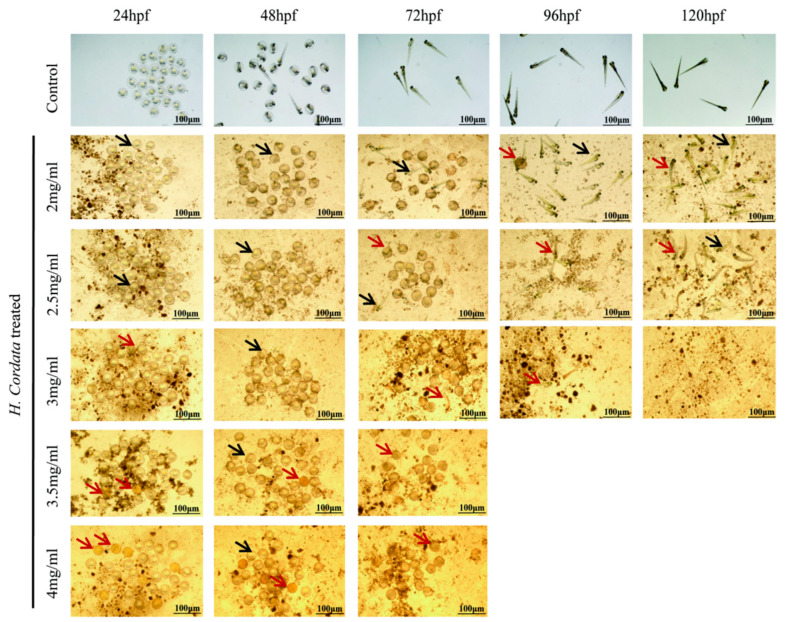
Cumulative survival rates of zebrafish embryos exposed to different concentrations (mg/mL) of *H. cordata* at 24, 48, 72, 96, and 120 hpf post-fertilization. Red arrows indicate dead zebrafish, black indicates surviving zebrafish.

**Figure 2 toxins-15-00073-f002:**
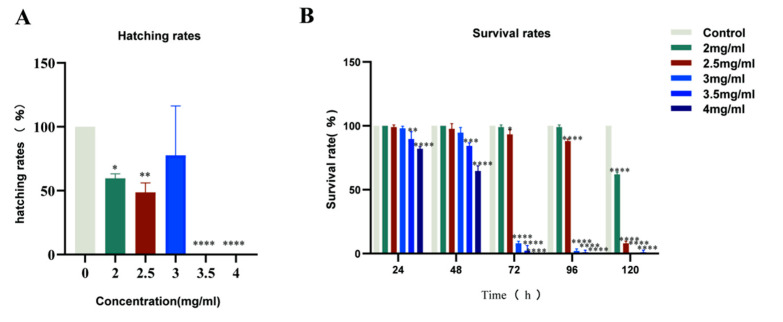
Effects of *H. cordata* on survival and hatching rates. (**A**) Hatching rates of zebrafish embryos exposed to different concentrations of *H. cordata* at 72 hpf. (**B**) Cumulative survival rates of zebrafish embryos exposed to different concentrations (mg/mL) of *H. cordata* at 24, 48, 72, 96 and 120 (hpf) post-fertilization.the results are presented as mean ± SE (*n* = 3). * *p* < 0.05, ** *p* < 0.01, *** *p* < 0.001 and **** *p* < 0.0001 versus control.

**Figure 3 toxins-15-00073-f003:**
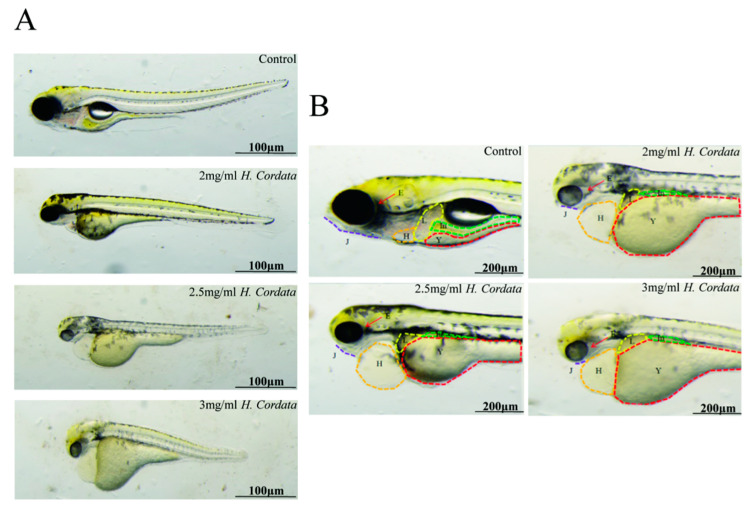
(**A**) Changes of body length of zebrafish treated with different concentrations of *H. cordata*. (**B**) Organ toxicity of zebrafish treated with Different concentrations of *H. cordata* Note: H = heart; J = jaw; In = intestines; L = the liver; Y = yolk sac; E = eyes.

**Figure 4 toxins-15-00073-f004:**
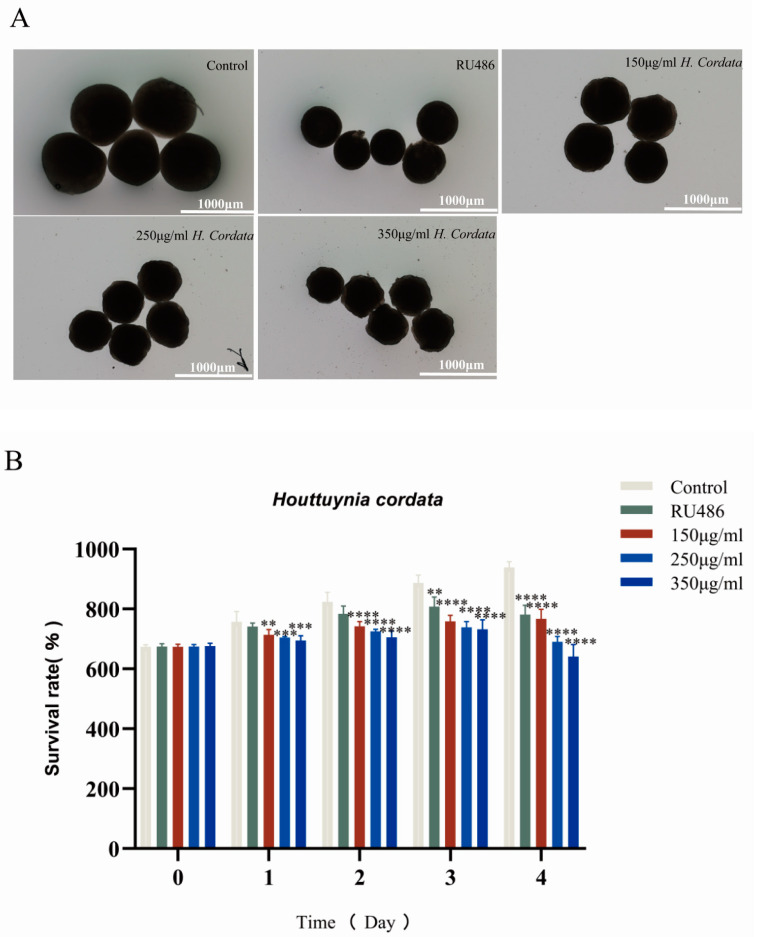
The changes of EBs’ diameters after exposure to *H. cordata*. (**A**) On day 4, the EBs’ diameters of the experimental groups were smaller than the control group. (**B**) M: The histogram and statistical analyses of EBs’ diameter at 0–4 days. *n* = 5, ?x ± s. ** *p* < 0.01, *** *p* < 0.001 and **** *p* < 0.0001 versus control.

**Figure 5 toxins-15-00073-f005:**
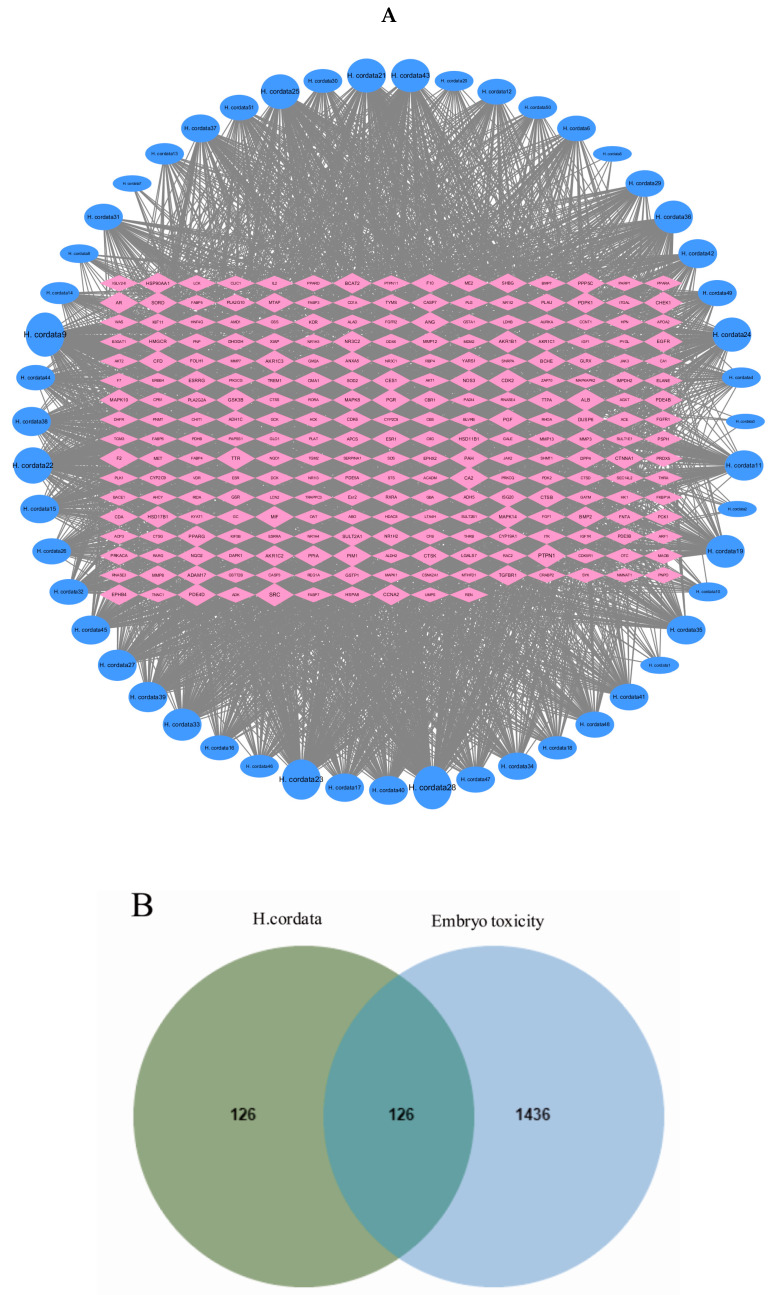
(**A**) The active components and target network of *H. cordata*. The blue circle represents the component of H. cordata; the red diamond represents the target; the size of the node represents the size of the node degree. (**B**) There are common targets between embryotoxicity and drugs. (**C**) PPI core network screening flow chart. the node size is proportional to its degree value. the larger the node, the more important the target in the network.

**Figure 6 toxins-15-00073-f006:**
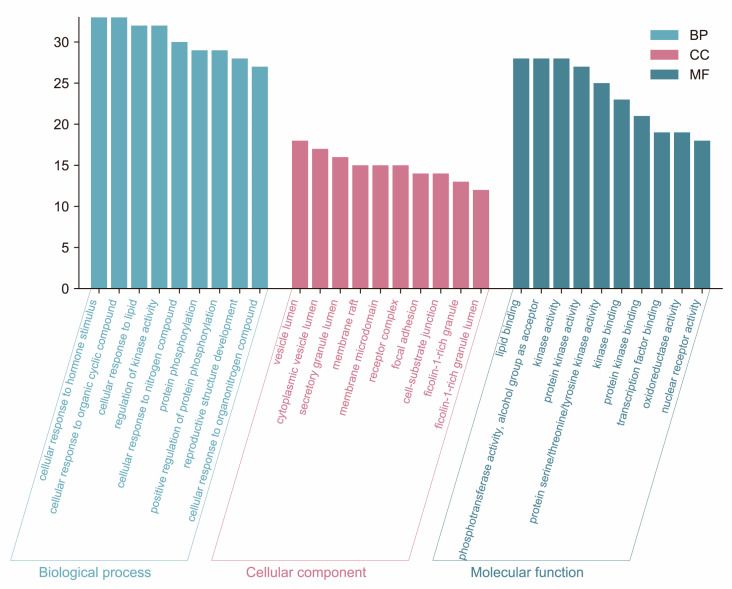
GO enrichment analysis of the 126 potential therapeutic targets. GO enrichment analysis: the top 10 biological processes, molecular functions, and cell components.

**Figure 7 toxins-15-00073-f007:**
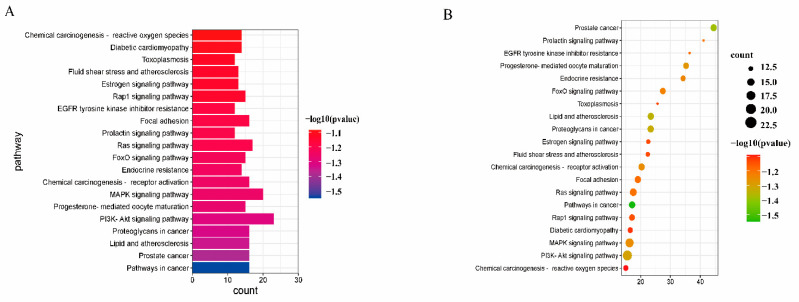
KEGG enrichment analysis of the 126 potential therapeutic targets. (**A**) Histogram of the top 20 KEGG pathways (**B**) Bubble plot of the top 20 KEGG pathways.

**Figure 8 toxins-15-00073-f008:**
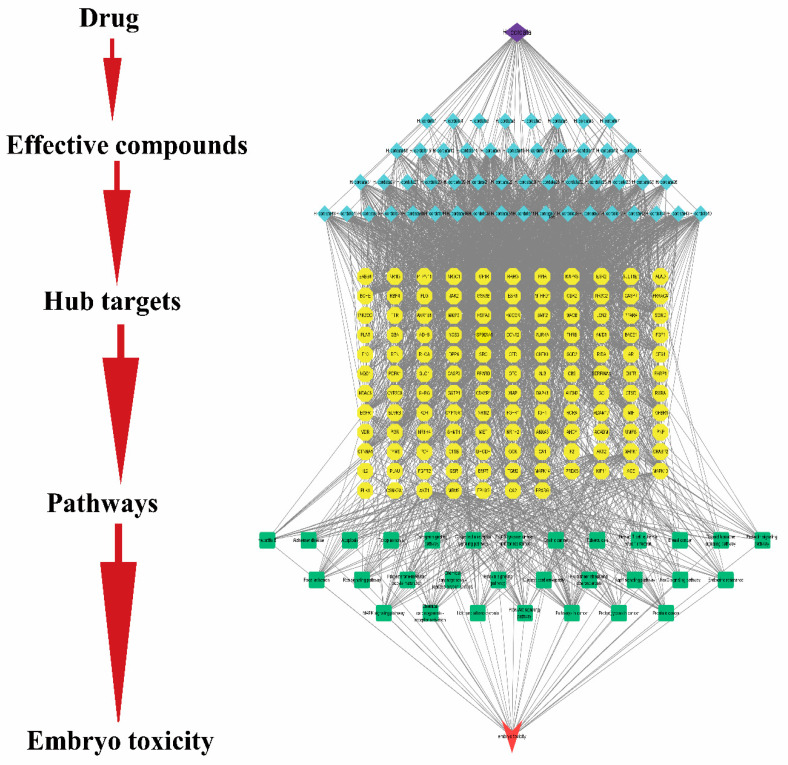
The network of compound-target-pathway of *H. cordata* for embryotoxicity. The network consists of 51 components, 126 common targets, and 30 pathways. The purple color represents herbs; blue indicates bioactive compounds from *H. cordata*; yellow stand for the target genes; green represents signal pathway; red is embryotoxic.

**Figure 9 toxins-15-00073-f009:**
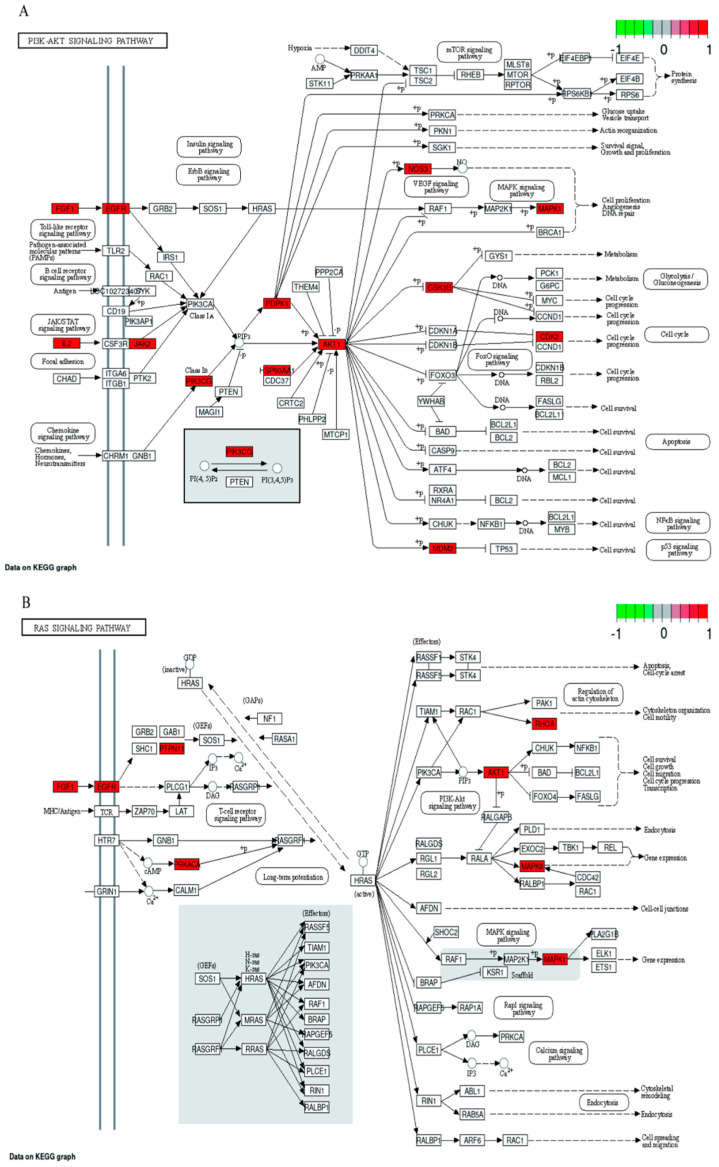
Schematic diagram of signaling pathways related to embryonic toxicity (**A**) PI3K-Akt signaling pathway, (**B**) Ras signaling pathways, (**C**) MAPK signaling pathway.

**Figure 10 toxins-15-00073-f010:**
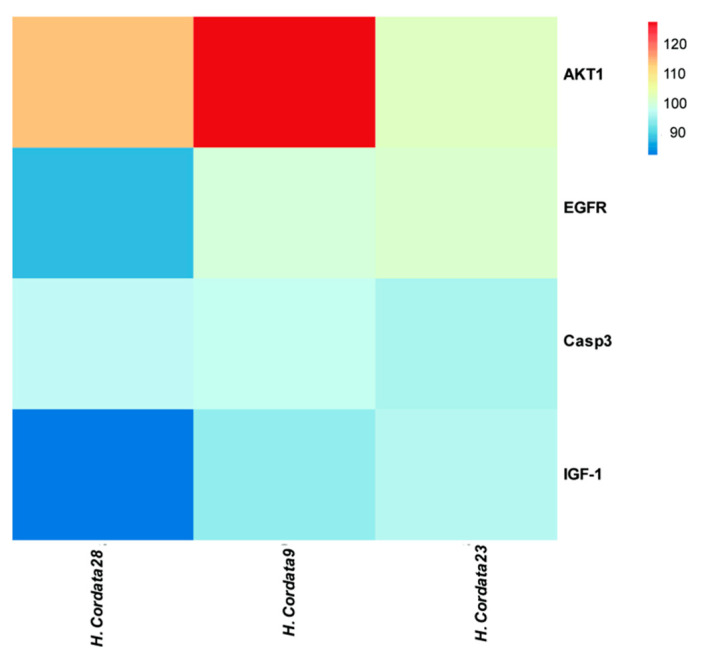
The binding energy of the main active components of *H. cordata* and core target genes.

**Figure 11 toxins-15-00073-f011:**
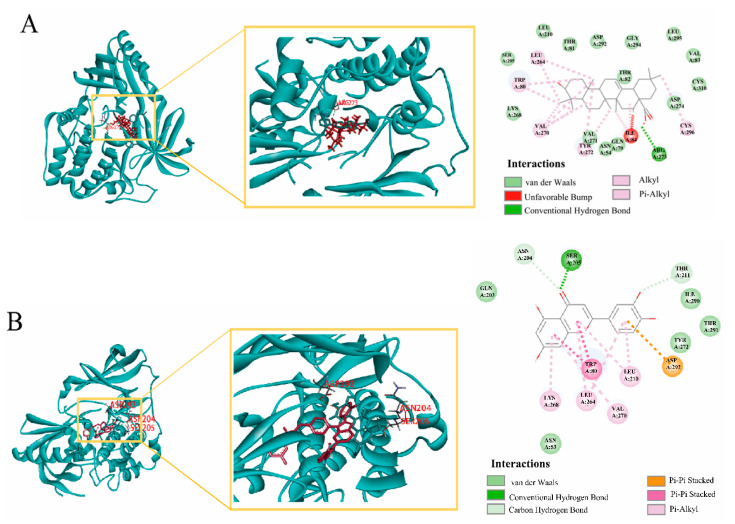
(**A**) Results of AKT1 docking with oleanolic acid, (**B**) Results of AKT1 docking with Luteolin, (**C**) Results of AKT1 docking with Aristolochia lactam AII.

**Figure 12 toxins-15-00073-f012:**
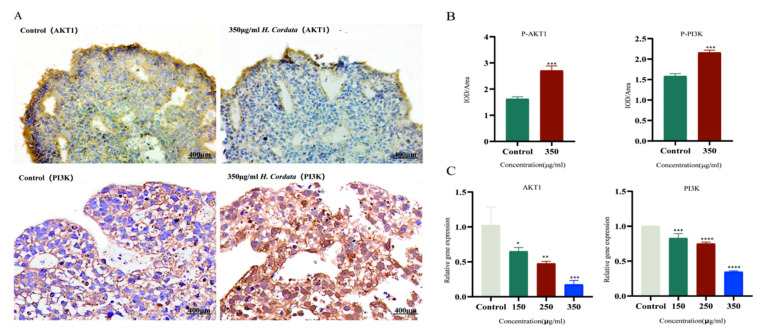
(**A**,**B**) Immunohistochemistry staining and statistical analysis of the protein levels of p-Akt and p-PI3K of EB treated with different concentrations of *H. cordata*. (**C**) Effect of different concentrations of *H. cordata* on the expression level of Akt1 and PI3K gene in EB. Data are expressed as mean ± standard deviation (*n* = 3). * *p* < 0.05, ** *p* < 0.01, *** *p* < 0.001 and **** *p* < 0.0001 versus control.

**Table 1 toxins-15-00073-t001:** Active compounds of *H. cordata*.

No.	Compounds
*H. cordata*1	strigol
*H. cordata*2	Sesamin
*H. cordata*3	apigenin
*H. cordata*4	Scutellarein
*H. cordata*5	Genistein
*H. cordata*6	pilloin
*H. cordata*7	Lauric acid
*H. cordata*8	Methyl ferulate
*H. cordata*9	Luteolin
*H. cordata*10	piperolactam B
*H. cordata*11	piperolactam C
*H. cordata*12	piperolactam D
*H. cordata*13	4-amino-N-(2-phenylethyl)benzamide
*H. cordata*14	asimilobine
*H. cordata*15	Norisoboldine
*H. cordata*16	laetanine
*H. cordata*17	Liriodenine
*H. cordata*18	N-Methylasimilobine
*H. cordata*19	perlolyrine
*H. cordata*20	Vomifoliol
*H. cordata*21	acantrifoside E
*H. cordata*22	Epigensexinmethyl ether
*H. cordata*23	Aristolochia lactam AII
*H. cordata*24	strigone
*H. cordata*25	Kaempferol
*H. cordata*26	sorgomol
*H. cordata*27	5-deoxystrigol
*H. cordata*28	oleanolic acid
*H. cordata*29	Phenyl β-D-glucopyranoside
*H. cordata*30	3-hydroxy-1,2-dimethoxy-5-methyl-5H-dibenzoindol-4-one
*H. cordata*31	3-methoxy-5-methyl-5H-benzodioxolo-benzoindol-4-one
*H. cordata*32	1,2,3,4,5-pentamethoxy-dibenzo-quinolin-7-one
*H. cordata*33	CARBOSTYRIL,4-HYDROXY-1-METHYL-3-[[P-(PHENYLAZO)PHENYL]AZO]-(7CI)
*H. cordata*34	7-oxodehydroasimilobine
*H. cordata*35	lysicamine
*H. cordata*36	atherospermidine
*H. cordata*37	3-methoxy-6H-benzodioxolo-benzoquinoline-4,5-dione
*H. cordata*38	3-methoxy-6-methyl-6H-benzodioxolo-benzoquinoline-4,5-dione
*H. cordata*	ouregidione
*H. cordata*40	cepharadioneB
*H. cordata*41	cepharadione A
*H. cordata*42	1,2,3-trimethoxy-4H,6H-dibenzoquinolin-5-one
*H. cordata*43	1,2,3-trimethoxy-6-methyl-4H,6H-dibenzoquinolin-5-one
*H. cordata*44	1,2-dimethoxy-3-hydroxy-5-oxonoraporphine
*H. cordata*45	1,2,3-trimethoxy-3-hydroxy-5-oxonoraporphine
*H. cordata*46	sauristolactam
*H. cordata*47	cepharanone B
*H. cordata*48	Aristolactam A
*H. cordata*49	Aristolactam B
*H. cordata*50	AristolactamBⅡ
*H. cordata*51	Isoramanone

**Table 2 toxins-15-00073-t002:** Molecular docking.

Core Components	Key Targets	Docking Score
oleanolic acid	AKT1	113.486
Luteolin	AKT1	126.616
Aristolochia lactam AII	AKT1	102.669
oleanolic acid	EGFR	89.5131
Luteolin	EGFR	100.632
Aristolochia lactam AII	EGFR	101.622
oleanolic acid	CASP3	97.9379
Luteolin	CASP3	98.2926
Aristolochia lactam AII	CASP3	96.8157
oleanolic acid	IGF-1	84.2265
Luteolin	IGF-1	95.4801
Aristolochia lactam AII	IGF-1	97.4921

## Data Availability

The data presented in this study are available in this article and Appendix A.

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
