# Peer review of "The Mechanism of Houttuynia cordata Embryotoxicity Was Explored in Combination with an Experimental Model and Network Pharmacology"

_toxins, 2023, doi:10.3390/toxins15010073_

Round 1

Reviewer 1 Report

Comments about the manuscript:

“The mechanism of Houttuynia cordata embryotoxicity was explored in combination with an experimental model and network pharmacology”.

Houttuynia cordata (H. cordata) is a commonly used herb both in food and in traditional Chinese medicine. But if its hepatotoxic effects are evaluated, its embryotoxicity by long-term exposure is not yet well known. Accordingly, the aim of this work was to investigate the effects of H. cordata and the mechanism of toxicity on embryonic development using several methods. For this, the authors evaluated the toxic effects of H. cordata on zebrafish embryos and human-induced pluripotent stem cells (hiPSCs) embryoid bodies administered by different principal active gents of H. cordata. The effects were demonstrated by RT-PCR and immunohistochemistry. The results of this study suggest that H.cordata may affect embryonic development by influencing the PI3K-Akt signaling pathway.

I find this work interesting and useful like any work which tends to confront a traditional pharmacopoeia with scientific data, in order to evaluate its effectiveness and/or its toxicity. This article could be published after some improvements to the manuscript.

Remarks

Given that the work requires experiments on animal embryos and uses human embryoid bodies, it is necessary to indicate in the article the references of the authorizations required to experiment. However, I did not find these references at the end of the manuscript. Please check if this has been given and, if not, add the references in the document.

Page 1, abstract, lines 10-17: the sentence is very long with several points. It would be better to divide it into several short sentence.

Page 2, beginning of line 54: delete one "in"

Page 2, figure 1: the pictures are very small and need a minimum of explanations.

Page 2, line 73: “Zebrafish embryos at 0 hpf were exposed to 2, 25, 3, 3.5, and 4 mg/ml”: replace “25” with “2.5”.

Page 3, figure 2: This figure needs some explanations. What is represented in each picture? Indicate with arrow the presence of zebrafish normal embryos, disformed embryos, what are the different structures observed on the pictures. Add a scale bar on each picture.

Page 4, lines 96 and 97, figure 3: use italics to write “H. cordata”.

Page 4, line 111, figure 4: add a scale bar on pictures.

Page 5, line 115, figure 5A: add a scale bar on pictures.

Page 5, lines 126-127. “A sum of 252 gene targets was acquired. As shown in Figure 6A”: I think it would better to write “A sum of 252 gene targets was acquired, as shown in Figure 6A”.

Page 6, line 143: write “respectively. The values” instead of “respectively. the values”.

Page 7, line 156: give in full here the significance of GO and KEGG for it is the first time these abbreviations appear in the text.

Page 8, figure 8: this figure is too small and hard to be read.

Page 9, lines 205: give in full here the significance of RMSD for it is the first time this abbreviation appears in the text. Delete this signification at line 213.

Page 11, line 245, figure 13. Use italics to write “H. cordata”.

Page 15, line 403, Embryoid body (EBs) assay. “hiPSCs were purchased from Beijing Saibei Biotechnology Company,”: specify in full the name of these embryoid bodies: “human-induced pluripotent stem cells”.

Page 17, line 486: “Select EBs (12 μm) in good shape to make ice slices.”: give some details about the preparation of "ice slices" (frozen section?). Were the embryoids preserved in fixative or not? If so, what fixer? Were they included in a resin? Thank you to specify.

How were the negative controls prepared? By omission of the first antibody? Otherwise?

Bibliographical references: check that the references are well presented according to the standards of the journal. Write the genus and species names in italics.

Figure S1: give a title at this figure.

Author Response

Jan. 7, 2023

Ms. Jolene You

Assistant Editor

Toxins

DearMs. Jolene You,

Thank you for your letter and the reviewers' comments on our manuscript entitled "The mechanism of Houttuynia cordata embryotoxicity was explored in combination with an experimental model and network pharmacology" (manuscript ID: toxins-2145961). These comments are valuable and helpful for us to revise and improve our paper, and they are also important for guiding our research. We have carefully studied these comments and made revisions for consideration for publication in Toxins.

In the revision, we made almost all suggested changes to the images and manuscript content, and fully addressed the comments made by the reviewers and the editor. "Track changes" were used in the paper, and the changes are marked in red.

All authors have read and approved the re-submission of the manuscript! If you have any questions, please let me know!

Thank you for your consideration of our paper and we are looking forward to hearing from you!

Sincerely yours,

Reviewer 2 Report

This article is covering some aspects of embriotoxicity of the Houttuynia cordata (perennial herb) on embryonic development of Zebrafish and based on the 2007 idea of network pharmacological analysis. 

The specific aims of this article are exclusively directed to examine active agents producing toxicity such as oleanolic acid, lignin and aristolactam. The mechanism of embriotoxicity of such agents is affecting the PI3K-Akt signaling pathway, MAPK signaling pathway, and Ras signaling pathway by modulating specific targets such as AKT1, EGFR, CASP3 and IGF-1. The immunochemistry results clearly showed that AKT1 inhibition and P13K expression were very significantly reduced in the embryoid body.  

The article is concluded with a collection of 45 mostly recent references. Additionally, all 13 Figures and 2 Tables are very informative and with concise important data comparison and references. Specifically, flowchart on fig 1. is clearly describing the idea of network pharmacology, and molecular docking helpful in providing safety assessment and information on the clinical use of H.cordata. This will constitute the important goals and novelty of this paper.

            The following suggested changes and recommendations should be introduced before the publication of the manuscript.

1.     Page 3. Line 95. Figure 3 should be modified with sequence  “B” should be “A” and “A” should be ‘B” by this new sequence pointing the concentration first and following with survival rate (time) the effect of embriotoxicity it will be very clear. 

2.     Page 6. Table 1. Should be in the form of two columns instead of four. In this way we have a continuous flow of information of 51 active components of H. cordata compiled. 

3.     Page 11, figure 12. For the clarity and comparison of docking AKT1 with oleanolic acid (A) and luteolin (B) docking with aristolochialactam (C) should be in the same green color not brown. In this way the comparative docking is more visible and confirmative!! 

4.     Page 14, Conclusion line 354 this sentence is short and not specific and should be expanded into the text to highlight the importance of in vivo and in vitro experiment preformed.

The manuscript is of good quality and importance and is sequentially written and edited in order to meet the standard for the articles published in Toxins. Thus, I certainly recommend it for publication after the correction of these suggested minor changes and recommendations. 

Author Response

(The authors gave the same response as above.)
